# The Role of MicroRNAs in Myocarditis—What Can We Learn from Clinical Trials?

**DOI:** 10.3390/ijms232416022

**Published:** 2022-12-16

**Authors:** Olga Grodzka, Grzegorz Procyk, Aleksandra Gąsecka

**Affiliations:** 1Department of Neurology, Medical University of Warsaw, Banacha 1A, 02-097 Warsaw, Poland; 21st Chair and Department of Cardiology, Medical University of Warsaw, Banacha 1A, 02-097 Warsaw, Poland; 3Doctoral School, Medical University of Warsaw, 02-091 Warsaw, Poland

**Keywords:** myocarditis, microRNA, cardiomyopathy, inflammation, clinical trials

## Abstract

Myocarditis is an inflammatory disease of the heart with a viral infection as the most common cause. It affects most commonly young adults. Although endomyocardial biopsy and cardiac magnetic resonance are used in the diagnosis, neither of them demonstrates all the required qualities. There is a clear need for a non-invasive, generally available diagnostic tool that will still remain highly specific and sensitive. These requirements could be possibly met by microribonucleic acids (miRNAs), which are small, non-coding RNA molecules that regulate many fundamental cell functions. They can be isolated from cells, tissues, or body fluids. Recently, several clinical studies have shown the deregulation of different miRNAs in myocarditis. The phase of the disease has also been evidenced to influence miRNA levels. These changes have been observed both in adult and pediatric patients. Some studies have revealed a correlation between the change in particular miRNA concentration and the degree of cardiac damage and inflammation. All of this indicates miRNAs as potential novel biomarkers in the diagnosis of myocarditis, as well as a prognostic tool for this condition. This review aims to summarize the current knowledge about the role of miRNAs in myocarditis based on the results of clinical studies.

## 1. Introduction

### 1.1. Basic Knowledge and Statistics about Myocarditis

According to the statement of the World Health Organization, myocarditis is defined as an inflammatory disease of the myocardium. It is most commonly caused by infections (primarily viral) and autoimmunization. In some cases, the etiology of the disease remains undetermined, and it is classified as an idiopathic form [1,2]. A histologic examination must be performed for the definite diagnosis of myocarditis. Viral myocarditis requires a polymerase chain reaction (PCR) detection of viruses, while autoimmune myocarditis is defined as myocarditis with a negative viral PCR test, regardless of the cardiac autoantibodies count [2,3]. Noteworthily, the disease usually affects young people, mostly aged between 20 and 50 years, not suffering from any other serious conditions [4]. These statistics are even more disturbing, taking into consideration the fact that the major long-term complication of myocarditis is dilated cardiomyopathy with chronic heart failure [5]. About 12–25% of patients die of myocarditis due to an end-stage cardiomyopathy [6]. However, in about 50% of cases, the inflammation of the heart might resolve without serious consequences, such as chronic cardiac dysfunction [7]. What is more, these different outcomes are usually seen regardless of the intensity of provided therapy. On the other hand, up to 10% of cardiomyopathies, initially unexplained, appear to be a result of myocarditis [8].

### 1.2. Available Diagnostic and Prognostic Tools in Myocarditis

Endomyocardial biopsy (EMB) remains the gold standard of myocarditis diagnosis, although it is an invasive procedure, and thus it is not a routine practice [9]. Concerning the experts’ consensus in the new statement, EMB is recommended for all patients with suspected myocarditis. However, many medical centers are not able to perform EMB procedures. Hence each patient with the suspicion of myocarditis should be referred to the hospital where EMB is performed [2]. The alternative method for myocarditis diagnosis is cardiac magnetic resonance (CMR), sometimes called a non-invasive gold standard [4]. The Lake Louise Criteria (LLC) are the recommended diagnostic criteria, although they may not have enough sensitivity in differentiation between recent and remote myocarditis [10,11]. New imaging techniques, such as T1 and T2 mapping, seem to be promising in terms of increasing the value of CMR [12]. Nevertheless, some limitations preclude implementing this method into the clinical routine, amongst them: the unavailability of advanced techniques, lack of validation in myocarditis, and some inconsistency with EMB findings [3,6].

The intensity of the treatment is not always positively correlated with the outcome of myocarditis [8]. The prognosis may be established based on EMB, which shows the underlying etiology and inflammation type [13]. Interestingly, fulminant myocarditis usually resolves with a better outcome than the chronic form of this disease [14]. Nonetheless, EMB remains an invasive procedure and is not always performed. Thus prognosis, in the case of many patients, stays unclear.

All the above implies the need for an efficient, easily accessible, and low-priced non-invasive diagnostic tool. Recently, much research has been focusing on investigating small particles, such as extracellular vesicles (EVs) or microribonucleic acids (miRNAs/miRs), circulating in the blood or found in tissues [15]. In terms of myocarditis diagnosis, miRNAs could meet the abovementioned expectations [16] (Figure 1).

### 1.3. Biosynthesis and Function of miRNAs

MiRNAs are a class of small ribonucleic acids (RNAs) containing from 19 to 25 nucleotides [17]. Although they are non-coding molecules, they play a crucial role in regulating gene expression [18]. A single miRNA molecule can target hundreds of different mRNAs associated with fundamental cell functions [19]. Approximately one-third of the regulation of the protein-coding genes is related to the miRNAs [20,21].

The initial step of the miRNA synthesis is the miRNA gene transcription by RNA polymerase II, forming a large primary miRNA (pri-miRNA) [22]. This molecule is composed of one or a few stem-loop structures and is subsequently cleaved by the Drosha enzyme into a single stem-loop structure called precursor miRNA (pre-miRNA) containing about 70 nucleotides [23]. Afterward, pre-miRNA is transported from the nucleus to the cytoplasm, where it is cleaved by a nuclease Dicer into a double-stranded miRNA consisting of approximately 22 nucleotides [24]. This miRNA is loaded into an argonaut protein to compose the RNA-induced silencing complex (RISC) [25]. The expression of the target mRNA is regulated by binding its 3′ end to the complementary 5′ end of the mature miRNA. Depending on the complementarity degree, mRNA can be repressed or degraded [21,24]. Furthermore, there are other mechanisms, independent of Drosha and Dicer, able to generate miRNAs. In addition, noncanonical binding sites have been identified [24,26].

It has been estimated that about 18% of miRNA-mRNA complexes are created with the 3′ end of miRNA instead of the 5′ end [27]. The information about involved miRNA’s ends was pointed out only by some of the authors of the analyzed literature, and therefore we have featured it only in some of the referred studies.

### 1.4. The Putative Role of miRNAs in Myocarditis

MiRNAs have already been proven to play a crucial role in the pathogenesis of different cardiovascular diseases, such as myocardial infarction, arrhythmias, heart failure, or coronary heart disease [28]. They have also been evidenced to be involved in the development of neoplastic diseases [29]. Therefore, current studies are investigating these small molecules with great interest [30,31,32,33,34]. MiRNAs can be isolated not only from cells and tissues but also from body fluids, such as cerebrospinal fluid or serum. Quantitative PCR (qPCR) is a method widely used to measure the level of miRNAs [35]. However, it can also be utilized to determine potential targets for studied miRNAs, which can be very useful in resolving different pathogenic pathways [17]. Several miRNA profiling studies have demonstrated that the blood and tissue levels of different miRNAs are significantly deregulated in myocarditis [36,37]. They might be used to differentiate from other cardiac disorders, to distinguish different stages of myocarditis, or to prognose the degree of cardiac damage and possible outcome.

The aim of this review is to summarize the current knowledge about the role of miRNAs in myocarditis based on past clinical studies [31].

## 2. MicroRNAs in Patients Suffering from Myocarditis

So far, few clinical studies investigating the role of microRNAs in myocarditis have been conducted. We searched PubMed Database and eventually included 21 original clinical research relevant to the discussed field. We have divided these studies into the following parts: (i) microRNAs in pediatric myocarditis patients, (ii) tissue microRNAs in adult myocarditis patients, and (iii) circulating microRNAs in adult myocarditis patients (Figure 2).

### 2.1. MicroRNAs in Pediatric Myocarditis Patients

Yan et al. studied a group of children suffering from viral myocarditis and compared them to healthy controls. The level of miR-146b was significantly higher in the first group as compared to the latter. The increase was positively correlated with the level of myocardial injury [38]. Another research group demonstrated that miR-217 and miR-543 levels were also significantly higher in children suffering from viral myocarditis compared to healthy controls. The results of this research suggest that the measurement of miRNAs level might be a novel diagnostic tool for myocarditis which is usually underdiagnosed in the pediatric population [39].

Goldberg et al. compared the levels of different miRNAs in children suffering from viral myocarditis in different phases of this disease. Two miRNAs demonstrated significant changes. The level of miR-208a was elevated in the acute phase, and it was subsequently diminishing. The decrease in the miR-21 level was observed during the resolution or chronic phase, while in the acute and subacute phases, it was increased. Additionally, another miR-208b expression level during the subacute phase was negatively correlated with systolic left ventricle function during the chronic phase, which suggests a prognostic value of the miR-208a measurements. Both miRNAs can be considered diagnostic biomarkers for myocarditis [40].

Zhang et al. conducted a study investigating children suffering from viral myocarditis. The level of miR-381 was measured in these patients, and it appeared to be decreased compared to its level in children that had recovered from myocarditis. Moreover, it was negatively correlated with the levels of cyclooxygenase-2, which is a substantial enzyme for inflammatory cytokines production [41]. Moreover, there was another study that included children suffering from viral myocarditis and compared them to healthy individuals. It was observed that the level of miR-133b was lower in the group of children with myocarditis. The severity of myocardial lesions was also assessed, and a negative correlation between miR-133b expression and heart damage in viral myocarditis was observed [42].

Interesting results were obtained by Wang et al., who analyzed the levels of miR-1 and miR-146b in children suffering from viral myocarditis, comparing them to healthy individuals. The upregulation of miR-146b and downregulation of miR-1 was observed in the study group. The correlation between these miRNAs’ levels and ejection fraction was also demonstrated: miR-1 was negatively correlated, while miR-146b was positively correlated. Thus, both miR-1 and miR-146b can be potential biomarkers in the diagnosis and prognosis of viral myocarditis [43]. All studies discussed in this subsection with additional data are summarized in Table 1.

### 2.2. Tissue MicroRNAs in Adult Myocarditis Patients

Besler et al. investigated patients suffering from viral myocarditis. The elevated levels of miR-133a and miR-155 were observed in these patients as compared to patients with dilated cardiomyopathy. Noteworthily, miR-133a and miR-155 levels were positively correlated with the inflammatory cell count in patients with viral myocarditis. Moreover, an increased level of miR-133a was associated with reduced fibrosis and myocytes’ necrosis, as well as the improved function of the left ventricle. Patients presenting with higher levels of miR-133a had increased overall survival and decreased occurrence of malignant arrhythmias. Moreover, they were less often hospitalized due to heart failure [44]. Slightly opposite results concerning miR-133a were obtained by Ferreira et al. They studied myocardial samples derived from patients suffering from Chagas disease (manifesting mainly with myocarditis). They discovered that miR-1, miR-133a, miR-133b, miR-208a, and miR-208b were downregulated in samples from patients with Chagas disease as compared to samples from healthy hearts of organ donors. Moreover, three of them: miR-1, miR-133a, and miR-208b, were significantly less expressed in the study group as compared not only to organ donors but also to patients with DCM. Some targets for these miRNAs were identified, and they were found to be responsible for cardiac hypertrophy and dysfunction or normal proliferation of cardiomyocytes [45].

Bao et al. investigated patients suffering from acute myocarditis and compared them to healthy controls. Eight different miRNAs were assessed, and it was found that miR-155 and miR-148a presented higher levels in heart tissue derived from patients with myocarditis [46]. Another research group investigated patients suffering from viral myocarditis, and they observed a downregulation of miR-214 and miR-146b in patients with myocarditis as compared to healthy controls [47].

Kühl et al. investigated patients with active or latent infection of Parvovirus B19 (involved in the pathogenesis of both myocarditis and DCM). The levels of different miRNAs were measured, and 29 miRNAs appeared to be differently expressed. Target genes for altered miRNAs were involved in cardiomyopathy-related pathways, inflammatory response, or mitochondrial energy metabolism [48]. Similar research was performed by Corsten et al., who investigated patients suffering from viral myocarditis. They observed the deregulation of 107 different miRNAs, of which 21 were upregulated, and 37 were downregulated by more than 1,5-fold. MiR-146b, miR-511, miR-155, miR-889 and miR-212 were the most upregulated miRNAs while miR-361, miR-106a and miR-93 were the most-downregulated ones [37]. All studies discussed in this subsection with additional data are summarized in Table 2.

### 2.3. Circulating MicroRNAs in Adult Myocarditis Patients

Blanco-Domínguez et al. investigated patients suffering from acute myocarditis, comparing them to patients with myocardial infarction (MI) and healthy controls. The first group of patients was shown to present increased levels of hsa-miR-Chr8:96 as compared to other groups. Moreover, the results remained significant after adjusting factors such as age, sex, ejection fraction, and troponin concentration in serum [49]. A similar study was performed by Nie et al., who investigated patients affected by fulminant myocarditis and compared them to patients suffering from MI and healthy controls. The level of miR-4281 was elevated in the group of patients with myocarditis and MI as compared to healthy controls, while the level of miR-4763-3p was significantly higher only in myocarditis patients. Interestingly the extent of demonstrated increase in miRNA levels was negatively correlated with the severity of fulminant myocarditis. Therefore, miR-4763-3p appears as a promising biomarker to differentiate myocarditis from MI [36].

In another study, Aleshcheva et al. included three groups: (i) myocarditis patients, (ii) dilated cardiomyopathy (DCM) patients, and (iii) healthy donors. After evaluation of different miRNA levels in serum within these groups, it was shown that Let-7f, miR-93, miR-197, miR-223, and miR-379 were significantly deregulated in patients suffering from myocarditis. The expression of miR-93, miR-197, and miR-379 was upregulated compared to both remaining groups, while let-7f and miR-223 levels were downregulated compared to healthy controls. The specificity of myocarditis identification with the use of the aforementioned biomarkers in a single serum sample was within the range of 93–95% [50].

Chen et al. investigated patients suffering from fulminant myocarditis, comparing them to healthy individuals. The increased levels of miR-29b and miR-125b in plasma were observed in the first group. Interestingly, this upregulation was positively correlated with the area of myocardial edema and was negatively correlated with the left ventricle ejection fraction. However, miR-29b demonstrated higher sensitivity and specificity for the fulminant myocarditis diagnosis than miR-125b [51]. Obradovic et al. investigated patients with the clinical suspicion of myocarditis, and after performing a biopsy, they divided them into two groups: (i) study group—myocarditis confirmed and (ii) control group—myocarditis excluded, diagnosis of dilated cardiomyopathy. The levels of six different miRNAs were measured, and it appeared that two of them were deregulated. To be more precise—the higher plasma levels of miR-155 and miR-206 were observed in the study group as compared to the control one [52].

Patients suffering from acute viral myocarditis were compared to healthy volunteers in another study. It was demonstrated that they had significantly increased levels of miR-21-5p and miR-1-3p as compared to the control group. Moreover, the upregulation was associated with myocardium damage and cardiac dysfunction [53]. Zhang et al. investigated a group of patients affected by fulminant myocarditis. The levels of several, i.e., miR-30a, miR-192, miR-146a, miR-155, and miR-320a, were evaluated and compared to the levels in patients with non-fulminant myocarditis and healthy controls. It was observed that patients with fulminant myocarditis presented higher levels of these miRNAs in comparison to other groups. The miRNA panel comprising miR-155 and miR-320a showed sufficiently high accuracy to distinguish between fulminant myocarditis and non-fulminant myocarditis. What is worth mentioning, this combination of miRNAs was shown to present greater diagnostic value than the levels of C-reactive protein and cardiac troponins, even when analyzed together [54].

Fan et al. studied the role of miRNAs among patients suffering from viral myocarditis. It was observed that five miRNAs were deregulated due to viral infection—patients presented increased levels of miR-181d, miR-30a, and miR-125a and decreased levels of miR-155 and miR-21. The study group was compared to patients presenting with unexplained ventricular tachyarrhythmias but without any markers of inflammation or impaired ejection fraction and with excluded viral infection [55]. Finally, it was demonstrated that the level of miR-98 was lowered in patients suffering from myocarditis as compared to healthy controls. The FAS and FASL genes (essential for apoptosis) were indicated as targets for miR-98. These results demonstrated the role of investigated miRNA in the inhibition of cardiomyocytes’ apoptosis [56]. All studies discussed in this subsection with additional data are summarized in Table 3.

## 3. Conclusions

Multiple studies concerning the role of miRNAs in myocarditis have been conducted so far. Nevertheless, the majority of them are preclinical studies, while there is relatively little clinical research. Therefore, in this review, we have summarized the latter. One-third of the included studies were investigating the pediatric population, which proved that not only young adults, who were the mainly affected group, but also children could be suffering from myocarditis. It implies an even higher demand for effective diagnostic and prognostic tools.

Different clinical studies demonstrated that the levels of multiple miRNAs were significantly upregulated or downregulated in both serum and tissue derived from patients suffering from myocarditis. Furthermore, observed deregulation was usually correlated with the degree of heart damage, inflammation, or even survival rate, which suggested the putative prognostic role of the miRNAs’ measurements. Importantly, miRNA levels in patients suffering from myocarditis were different from those measured in patients with other cardiac diseases, such as myocardial infarction or non-inflammatory dilated cardiomyopathy, which makes them promising diagnostic biomarkers. It was also proved that particular miRNAs can be used to distinguish between different phases of myocarditis. It might be utilized to identify patients in the fulminant or acute phase of the disease, most often requiring even more intensive care (Figure 3).

## 4. Future Perspectives

Undoubtedly, more studies investigating the role of miRNAs in cardiac diseases, including myocarditis, are highly warranted. Particularly, clinical studies assessing miRNAs as potential diagnostic and prognostic biomarkers are highly desirable. According to the available literature, miRNAs may play an important role both in the primary diagnosis of myocarditis as well as in the differential diagnosis. Taking into consideration that prognosis in myocarditis is usually unclear and that miRNAs were shown to be correlated with the severity of the disease—further research evaluating the potential use of miRNAs as the prognostic tool is indicated.

There are compelling arguments endorsing the use of miRNAs as biomarkers, such as their non-invasiveness or relatively short time of waiting for the measurement results. All the above suggests that shortly there might be even more interest in miRNAs, which hopefully will result in a breakthrough in myocarditis management in clinical practice.

Importantly, particular miRNAs might be proven not only to correlate with myocarditis severity but also to exert a causal effect on the development and course of the disease. Such a discovery could lead to new treatment regimens aiming to adjust these particular miRNA levels. Moreover, miRNAs might potentially be used as therapeutics *per se* to regulate the expression of genes involved in disease progression.

## Figures and Tables

**Figure 1 ijms-23-16022-f001:**
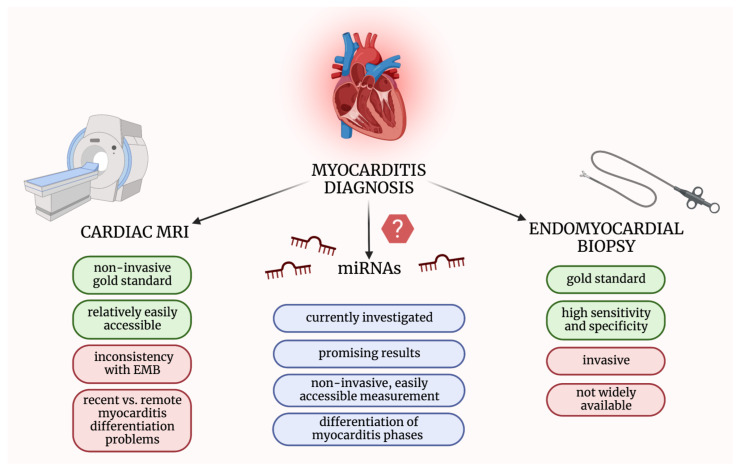
Comparison of methods used in myocarditis diagnosis and microRNAs’ advantages as potential diagnostic biomarkers. EMB—endomyocardial biopsy; miRNAs—micro-ribonucleic acids; MRI—magnetic resonance imaging.

**Figure 2 ijms-23-16022-f002:**
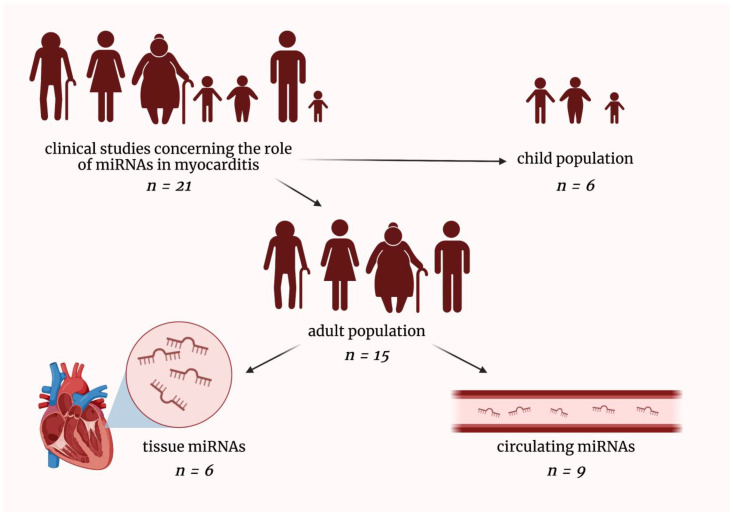
Graphical presentation of dividing clinical studies included in this review. miRNAs—micro-ribonucleic acids; *n*—a number of included clinical studies in a given field.

**Figure 3 ijms-23-16022-f003:**
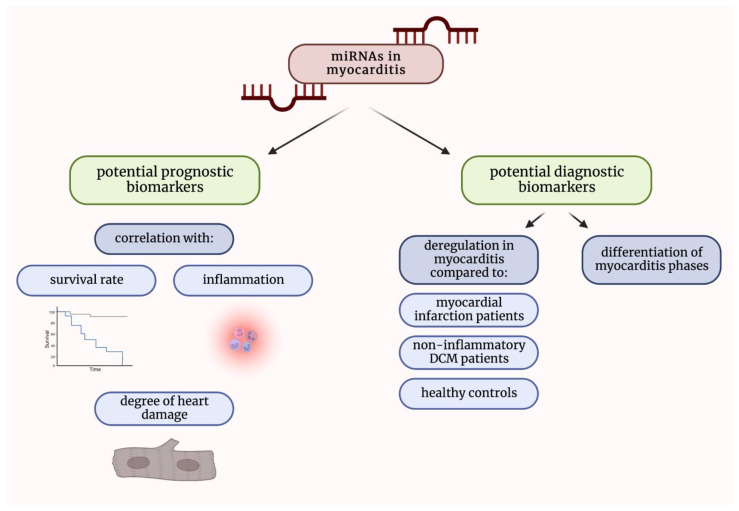
Graphical summarization of the role of miRNAs as potential diagnostic and prognostic biomarkers in myocarditis. DCM—dilated cardiomyopathy; miRNAs—micro-ribonucleic acids.

**Table 1 ijms-23-16022-t001:** Summary of recent studies regarding microRNAs in pediatric myocarditis patients.

Ref.	Year	Population	Comparison	miRNA	Outcome	Methodology
Yan et al. [38]	2021	48 VMC pediatric pts	40 HCs	miR-146b	↑ miR-146b in VMC pts compared to HCs	miRs in serum by qPCR
Xia et al. [39]	2020	30 VMC pediatric pts	HCs	miR-217 miR-543	↑ miR-217 and miR-543 in study groups compared to controls	miRs in blood by qPCR
Goldberg et al. [40]	2018	8 VMC pediatric pts	comparison of acute, subacute and resolution/chronic phase of VMC	miR-21 miR-208a	↑ miR-208a in acute phase compared to other phases ↓ miR-21 in resolution/chronic phase compared to other phases	miRs in serum by qPCR
Zhang et al. [41]	2018	26 VMC pediatric pts	33 pediatric pts recovered from VMC	miR-381	↓ miR-381 in VMC pts compared to recovered pts	miRs in serum by qPCR
Zhang et al. [42]	2017	36 VMC pediatric pts	HCs	miR-133b	↓ miR-133b in VMC pts	miRs in blood by qPCR
Wang et al. [43]	2016	119 VMC pediatric pts	120 HCs	miR-1 miR-146b	↓ miR-1 and ↑ miR-146b in VMC pts	miRs in serum by qPCR

↑—increased, ↓—decreased, HCs—healthy controls, miR/miRNA—microRNA, pts—patients, qPCR—quantitative polymerase chain reaction, ref.—reference, RNA—ribonucleic acid, VMC—viral myocarditis.

**Table 2 ijms-23-16022-t002:** Summary of recent studies regarding tissue microRNAs in adult myocarditis patients.

Ref.	Year	Population	Comparison	miRNA	Outcome	Methodology
Besler et al. [44]	2016	76 VMC pts	22 DCM pts	miR-133a miR-155	↑ miR-133a, miR-155 in MC pts ↑ miR-133a associated with ↓ fibrosis and ↑ LV function miR-133a and miR-155 levels correlated with inflammatory cells count in MC pts	miRs in heart tissue by qPCR fibrosis, inflammatory cells by HP LV function in ECHO
Ferreira et al. [45]	2014	10 CCC pts	6 DCM pts 4 HCs	miR-1 miR-133a miR-133b miR-208a miR-208b	↓ miR-1, miR-133a and miR-208b in CCC pts compared to others ↓ miR-133b and miR-208a in CCC pts compared to HCs	miRs in heart tissue by qPCR
Bao et al. [46]	2014	6 acute MC pts	6 HCs	miR-155 miR-148a	↑ miR-155 and miR-148a in MC pts	miRs in heart biopsy by qPCR
Chen et al. [47]	2015	8 acute VMC pts	8 HCs	miR-214 miR-146b	↓ miR-214 and miR-146b in VMC pts	miRs in heart tissue by qPCR
Kühl et al. [48]	2012	15 active B19V infection pts	45 inactive B19V infection pts	miRNAome	29 differently-expressed miRs predicted miR-Ts involved in immune response and energy metabolism	miRs in heart tissue by qPCR
Corsten et al. [37]	2012	4 acute MC pts	6 HCs	miRNAome	107 deregulated miRs in MC pts compared to HCs	miRs in heart biopsy by microarray analysis

↑—increased, ↓—decreased, B19V—Parvovirus B19, CCC—chronic Chagas disease cardiomyopathy, DCM—dilated cardiomyopathy, ECHO—echocardiography, HCs—healthy controls, HP—histopathology, LV—left ventricle, MC—myocarditis, miR/miRNA—microRNA, miR-Ts—miRNA target sequences, pts—patients, qPCR—quantitative polymerase chain reaction, ref.—reference, RNA—ribonucleic acid, VMC—viral myocarditis.

**Table 3 ijms-23-16022-t003:** Summary of recent studies regarding circulating microRNAs in adult myocarditis patients.

Ref.	Year	Population	Comparison	miRNA	Outcome	Methodology
Blanco-Domínguez et al. [49]	2021	39 acute MC pts	39 STEMI pts 38 NSTEMI pts31 HCs	miR-Chr8:96	↑ miR-Chr8:96 in MC pts compared to other groups	miRs in plasma by qPCR
Nie et al. [36]	2020	20 fulminant MC pts	5 MI pts 10 HCs	miR-4763-3p miR-4281	↑ miR-4763-3p in fulminant MC pts compared to other groups↑ miR-4281 in fulminant MC pts compared to HCs	miRs in plasma by microarray and qPCR
Aleshcheva et al. [50]	2020	343 AMC or VMC pts	71 DCM pts 85 HCs	Let-7f miR-197 miR-223 miR-93 miR-379	deregulation of these miRNAs only in MC pts: ↑ miR-93, miR-197, miR-379 compared to other groups ↓ let7f, miR-223 compared to HCs	miRs in serum by qPCR
Chen et al. [51]	2022	fulminant MC pts	HCs	miR-29b miR-125b	↑ miR-29b and miR-125b in fulminant MC patients compared to HCs	miRs in plasma by microarray analyses and qPCR
Obradovic et al. [52]	2021	60 MC pts	29 DCM pts	miR-155 miR-206	↑ miR-155 and miR-206 in MC pts compared to control group	miRs in plasma by qPCR
Marketou et al. [53]	2021	40 acute VMC pts	29 HCs	miR-21-5p miR-1-3p	↑ miR-21-5p and miR-1-3p compared to control group	miRs in blood by qPCR
Zhang et al. [54]	2021	99 fulminant MC pts	32 non-fulminant MC pts 105 HCs	miR-30a miR-192 miR-146a miR-155 miR-320a	↑ miR-30a, miR-192, miR-146a, miR-155 and miR-320a in fulminant MC pts compared to other groups	exosomal miRs by NGS and qPCR
Fan et al. [55]	2019	23 acute MC pts	12 HCs	miR-181d miR-30a miR-125a miR-155 miR-21	↑ miR-181d, miR-30a, miR-125a and ↓ miR-125a, miR-21 in MC pts compared to HCs	miRs in serum exosomes by qPCR
Zhang et al. [56]	2016	50 MC pts	50 HCs	miR-98	↓ miR-98 in MC pts	miRs in plasma by qPCR

↑—increased, ↓—decreased, AMC—autoimmune myocarditis, DCM—dilated cardiomyopathy, ECHO—echocardiography, HCs—healthy controls, HP—histopathology, LV—left ventricle, MC—myocarditis, MI—myocardial infarction, miR/miRNA—microRNA, miR-Ts—miRNA target sequences, NGS—next-generation sequencing, NSTEMI—non-ST-elevation myocardial infarction, pts—patients, qPCR—quantitative polymerase chain reaction, ref.—reference, RNA—ribonucleic acid, STEMI—ST-elevation myocardial infarction, VMC—viral myocarditis.

## Data Availability

Not applicable.

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
