# Peer review of "The Role of MicroRNAs in Myocarditis—What Can We Learn from Clinical Trials?"

_ijms, 2022, doi:10.3390/ijms232416022_

Round 1
Reviewer 1 Report
The manuscript is well written and very interesting. miRNAs represent the new frontier in medicine. MiRNAs are studied in various fields of medicine.
In addition to highlighting the role of miRNA biomarkers in myocarditis, the authors should also discuss their role in therapy.
What role will miRNAs play in myocarditis therapy?
I also recommend quoting these two articles 10.3390/diagnostics11010032
10.1038/s41598-019-56682-7
Reviewer 2 Report
This is well presented, well organised and timely to the point review. The figures are supporting the text.
Reviewer 3 Report
Thank you very much for inviting me to review an interesting article by Grodzka O. and colleagues. The authors analyzed role of Micro RNAs in myocarditis.
Myocarditis is caused by infection, toxicity and autoimmune response that may evolve to heart failure as a dilated cardiomyopathy..
Myocarditis is also diagnosed by gold standard examination such as Endomyocardial biopsy ( EMB) which is an invasive procedure or cardiovascular magnetic resonance imaging . Authors shows that recent advancement in molecular biology have increased our knowledge of myocarditis and may become an important biomarkers ( micro RNA) in diagnosis of acute myocarditis .
Authors introduce clinical studies both in pediatric and adult patients and divided them into 3 parts:
- micorRNA in pediatrics myocarditis patients
- circulating micro-RNA in adult myocarditis patients
- tissue microRNA in adult myocarditis patients
-
The discussion describes the significance of findings and shows that diagnosis of myocarditis remains a challenge because of the lack of methods which could be both specific, sensitive and also easily accessible. Author’s describes that several micro RNA studies have been conducted and the importance of that biomarkers has been recognized. For the future, we can also profile patients with different conditions to find also different expression of miRNAs at acute or chronic myocarditis. However miRNAs shows new insights into understanding of myocarditis and become promising tool in diagnosis oh that disease.
